# Implants for HIV prevention in young women: Provider perceptions and lessons learned from contraceptive implant provision

**Hilton Humphries** [ORCID]*, **Michele Upfold, Gethwana Mahlase, Makhosazana Mdladla, Tanuja N. Gengiah** [ORCID]**, Quarraisha Abdool Karim**

Centre for the AIDS Programme of Research in South Africa (CAPRISA), Nelson R Mandela School of Medicine, University of KwaZulu-Natal, Durban, South Africa

* Hilton.humphries@caprisa.org, hilthumphries@gmail.com

**Data Availability Statement:** The data that support the findings of this study are available from CAPRISA but restrictions apply to the availability of

## Abstract

Preventing new HIV infections, especially amongst young women, is key to ending the HIV epidemic especially in sub-Saharan Africa. Potent antiretroviral (ARV) drugs used as pre-exposure prophylaxis (PrEP) are currently being formulated as long-acting implantable devices, or nanosuspension injectables that release drug at a sustained rate providing protection from acquiring HIV. PrEP as implants (PrEP Implants) offers an innovative and novel approach, expanding the HIV prevention toolbox. Feedback from providers and future users in the early clinical product development stages may identify modifiable characteristics which can improve acceptability and uptake of new technologies. Healthcare workers (HCWs) perspectives and lessons learned during the rollout of contraceptive implants will allow us to understand what factors may impact the roll-out of PrEP implants. We conducted eighteen interviews with HCWs (9 Nurses and 9 Community Healthcare Workers) in rural KwaZulu-Natal, South Africa. HCWs listed the long-acting nature of the contraceptive implant as a key benefit, helping to overcome healthcare system barriers like heavy workloads and understaffing. However, challenges like side effects, migration of the implant, stakeholder buy-in and inconsistent training on insertion and removal hampered the roll-out of the contraceptive implant. For PrEP implants, HCWs preferred long-acting products that were palpable and biodegradable. Our findings highlighted that the characteristics of PrEP implants that are perceived to be beneficial by HCWs may not align with that of potential users, potentially impacting the acceptability and uptake of PrEP implants. Further our data highlight the need for sustained and multi-pronged approaches to training HCWs and introducing new health technologies into communities. Finding a balance between the needs of HCWs that accommodate their heavy workloads, limited resources at points of delivery of care and the needs and preferences of potential users need to be carefully considered in the development of PrEP implants.

these data, which were used under license for the current study, and so are not publicly available. Data are however available upon reasonable request from Moise Majyambere (Knowledge Management for CAPRISA, email: Moise. Majyambere@caprisa.org) and with permission of CAPRISA (data request form available at https:// www.caprisa.org/).

**Funding:** Funding: Funding for the study was provided by South African Medical Research Council Special initiative grant (00251), and partial support from the Department of Science and Innovation-National Research Foundation Centre of Excellence in HIV Prevention. The funders had no role in study design, data collection and analysis, decision to publish, or preparation of the manuscript.

**Competing interests:** The authors have declared that no competing interests exist.

## Introduction

Preventing new HIV infections remains a global challenge and a major barrier to achieving the United Nations 2030 goal of ending AIDS as a public health threat. Recent developments such as oral pre-exposure (PrEP) and the vaginal ring provide options that, for the first time, offer a cis-gender women-initiated HIV prevention method [1–4]. While these provide valuable protection against HIV infection, some young women aged 15–24 years old in sub-Saharan Africa (where 25% of global new infections occur) face challenges with adherence [5]. The PrEP landscape is rapidly expanding to address these adherence challenges by developing different long-acting slow-release products [6]. The array of new PrEP products under clinical development mirrors available biomedical family planning methods providing an opportunity to gain insights on provider and target population product preferences based on provider and user experiences of contraceptive options currently being used.

The 3-year, single rod etonogestrel contraceptive implant, Implanon NXT®, has been available through the public and private health care system in South Africa since 2014 [7–13]. Initial high uptake through public health care clinics was observed [7] but myths about the implant [14,15], systemic side effects, low levels of user knowledge, inadequately trained providers and inadequate community engagement contributed to a steady decline in the uptake and continued use of Implanon NXT [7–13]. Lessons and attitudes towards the contraceptive implant may offer insight into the design and roll-out of subdermal implants for HIV prevention. Antiretroviral drugs used as pre-exposure prophylaxis (PrEP) are currently in development to be formulated into subdermal implants, either routinely removable or biodegradable, or injectable nanosuspensions that release anti-HIV drugs at a sustained rate and provide protection from acquiring HIV. Thus, PrEP as an implant (PrEP Implants) offers a new technology to expand the HIV prevention toolbox. An implant containing tenofovir alafenamide (TAF) for PrEP HIV prevention is currently in the early stages of clinical development [16].

Challenges with the acceptability of new HIV prevention technologies mean that end-user and future-provider research at the early stages of clinical product development are critical. Health care workers (HCWs) act as important gatekeepers to the promotion of new health technologies, providing a critical view of the challenges to the uptake and implementation of new technologies within the public health care setting [17]. Previous research in urban/peri-urban areas of South Africa highlighted that the design and duration of coverage of PrEP implants were important to HCWs especially for overcoming barriers within health systems like staff shortages and capacity development [17]. As with potential users, when HCWs are engaged as part of the development process about modifiable aspects of new products (those characteristics which can be changed or modified) they may be more likely to promote them when they are licensed and available programmatically [18]. Additionally, HCWs can be engaged to contribute to improvements in messaging, information and modifiable aspects of the PrEP implants in a manner that is responsive to the needs of future users and improves the provider experience.

Engaging potential users and service providers ensures that the acceptability of products responds to the social and contextual factors in which they will be provided and used. This moves us beyond a product efficacy focus to developing products that fit practically into the context and lives of those who will use or provide them. This manuscript describes the perspectives of HCWs providing contraceptive implants or information about contraceptive implants in rural KwaZulu-Natal. We explore their insights into the design and contextual issues that may act as challenges and facilitators to the use of PrEP implants for HIV prevention amongst young women living in a high HIV burden rural community in South Africa.

## Materials and methods

### Study community

Vulindlela is a rural community situated 150km west of Durban, KwaZulu-Natal, South Africa. It has a high burden of HIV with an estimated antenatal prevalence of over 40% in women aged 20–24 years [19], and a population prevalence of 36.7% [20]. Access to health care is through nine primary health care clinics (PHCs). Outside of these PHC clinics, mobile clinics and community health care workers (CHWs) provide additional health care support within the community.

### Procedure

Data were collected between October 2019 and March 2020. Purposive sampling was used to identify a professional nurse and a CHW at each of the nine PHC clinics in the study community. To be eligible for the study, the participant had to be a professional nurse or CHW working at a primary health care clinic, and have self-reported experience with providing information about Implanon NXT (CHWs and nurses) or contraceptive services (nurses) which included Implanon NXT. There was no upper age limit or gender criteria for participating in the study. We included CHWs as they provided an important link between the formal health care system and the community [18]. To recruit participants, our CAPRISA community team approached each of the clinics and asked the clinic to identify the nurses and CHWs responsible for providing or doing community outreach with regards to family planning services at the clinic. Those who met the eligibility requirements were asked to participate in the study. All nurses approached, who were eligble, were willing to participate.

 A participatory approach was used to develop the in-depth interviews (IDI) guide that included discussions with local community partners including the CAPRISA Research Support Group. Following written informed consent, a short interviewer-administered survey was completed to capture basic demographics, experience with Implanon, and qualification, after which individual in-depth interviews (IDI) using a semi-structured interview guide were completed at a time convenient to the participant. All interviews were conducted by an experienced masters-level, female interviewer (GM), who is a trained nurse and who is fluent in English and isiZulu, the interviewer verbally translated the guides during the interview. The interviewer did not have a previous relationship with the participants, and explained the purpose of the research. The IDI guides (Nurse and CHW Interview Guides in S1 Appendix) included items to ascertain participant knowledge and experiences with the contraceptive implant; challenges and successes of past and present provision of the contraceptive implant; their views on the potential of implants for HIV prevention; and how they would introduce it to potential users and/or create demand and facilitate uptake in young women. To minimise disruption, interviews were conducted in a private space at the PHC clinic, and each interview lasted approximately one hour. All IDIs were conducted in isiZulu and were audio-recorded, transcribed and then translated into English. English transcripts were quality checked by a research assistant fluent in isiZulu and English against the isiZulu interview recordings for accuracy. Participants were offered refreshments but no reimbursement for their time or travel. Approval for the study was obtained from the University of KwaZulu-Natal Biomedical Research Ethics Committee (BE477/19) and the South African National Department of Health Ethics Committee (KZ_201910_017).

### Data analysis

An iterative process was used to analyse the data. After conducting a literature review and knowledge from the study team's prior experience integrating Tenofovir gel into family

planning services [21], preliminary themes were included in a draft codebook *a priori*. These were then validated by reviewing a sub-set of transcripts [22]. Through multiple independent reading and re-reading of a sub-set of transcripts, themes and codes were reviewed until consensus about the representativeness of the theme was reached by the primary research team. The themes identified were documented following guidelines by Boyatzis [23]. In line with a grounded theory approach, the consensus codebook was then applied to the full data iteratively to identify any new themes that needed to be added [24]. Once the coding framework was finalised, coding was undertaken using NVivo-12 (QSR International). Field notes taken by the interviewer were consulted to provide additional insight into the themes identified. This final coding was done independently by four members of the research team, followed by group meetings to reach a consensus and saturation on themes and codes and continued improvements on inter-rater reliability and adaptations to the codebook. Once inter-rater testing was complete, the themes were used as a basis for analysis and discussion of the overall research question.

## Results

### Demographics

A total of eighteen interviews were completed, nine with PHC nurses and nine with CHWs. All the participants were female (Note that in this study, women are defined as cis-gender women, i.e. individuals assigned female sex at birth and that identify their gender as a woman).The mean age of the PHC nurses was 41 years (range: 32–48 years), and 41 years for CHWs (range: 30–49 years). All those interviewed had experience providing information about Implanon NXT and discussing contraception with potential users. Details about the experience with inserting and removing Implanon NXT and Implanon NXT insertion/ removal training for PHC nurses are detailed in Table 1.

The analysis identified four major themes, 1) Benefits and challenges associated with contraceptive implants, 2) Important characteristics for an HIV prevention implant, 3) Capacity development and training of service providers, and 4) Engaging the local community on new technologies (Table 2). For the extracts, note the following conventions: I = interviewer, N = Nurse and C = Community Healthcare Worker.

### Benefits and challenges of the contraceptive implant that impact acceptability

As the characteristics of the contraceptive implant provide important insight into the factors that may similarly impact PrEP implants, this theme looks at the HCW experience of benefits and challenges of the contraceptive implants and how these may have impacted acceptability amongst young women. Sub-themes include Long-acting, Side effects, Migration of the Contraceptive Implant and the Social Context: Important Gatekeepers. These are discussed below:

**Long-acting nature of the contraceptive implant.** Across nurses and CHWs, the long-acting nature of the contraceptive implant was provided as its primary benefit. This was because 1) it did not rely on frequent user adherence (daily or 2 to 3 monthly), 2) less frequent provision meant less frequent pain, and 3) long-acting meant more user convenience, control, confidentiality and the benefits of saving the user both time and money as illustrated in the three quotes below:

*I see a lot of benefits, mainly time-saving, as I don't need to [come] to the clinic all the time. It saves both time and money. Also, if I feel pain I would feel it once and the next time I feel pain*

**Table 1. Demographic and experience details for nurses and CHWs working in Rural KwaZulu-Natal.**

|  | Nurses (N = 9) | CHW (N = 9) |
|---|---|---|
| **Mean Age (Age Range)** | **41 years (32–48 years)** | **41 years (30–49 years)** |
| **Mean years experience working as a CHW** |  | **11.6 years (Range: 4–19 years)** |
| **Nurses professional Experience with Implanon NXT Provision** |  |  |
| **Years' experience# with Implanon NXT** | **(N = 9)** |  |
| <2 | 11.0% (1) |  |
| 3–5 | 55.6% (5) |  |
| >5 | 33.3% (3) |  |
| Range | 1–9 years |  |
| **Number Implanon NXT inserted** | **(N = 9)** |  |
| 0 | 11% (1) |  |
| 1–10 | 22.2% (2) |  |
| >10–100 | 33.3% (3) |  |
| >100 | 33.3% (3) |  |
| **Number Implanon NXT removed** | **(N = 9)** |  |
| 0 | 11% (1) |  |
| 1–10 | 44.4% (4) |  |
| >10–100 | 44.4% (4) |  |
| >100 | 0% (0) |  |
| **Training on Implanon NXT** | **(N = 9)** |  |
| No training | 11% (1) |  |
| Theoretical | 11% (1) |  |
| Theoretical and practical | 44.4% (4) |  |
| Trained by a colleague | 33.3% (3) |  |
| **Received Supervision and mentorship** | **(N = 9)** |  |
| Yes | 55.6% (5) |  |
| No | 33.3% (3) |  |
| Not applicable | 11% (1) |  |

#: Experiences are defined as the years in which the provider has been trained to provide information or services related to Implanon.

*will be after three years when the implant is removed unlike feeling pain every 3 months. (IDI3_Nurse)*

*I: Okay. Okay. In your contraceptive counselling, eh. . .what do you cover?*

*N: Okay, firstly, I cover the advantages because one of the advantages of the implanon, is that it stays long. We are using that three-year one, so that is the advantage is that they do not have to come to the clinic every three months to access family planning. Another advantage is that if you feel like falling pregnant you can come to the clinic and remove it and go back to your fertility. The other advantage is that eh. . .there is no need to tell your partner about it (IDI16_Nurse)*

*I: Okay. What do you like about the implant?*

*C: It stays for a long time. It's unlike the injection that you have to inject every 2 months. It stays for up to 3 years. And you can replace it when it time has lapsed. (IDI11_CHW)*

**Table 2. Outline of major themes identified during analysis.**

| Major Theme | Sub-Themes |
|---|---|
| Benefits and Challenges of the Contraceptive Implant that Impact Acceptability | Long-acting: Important theme amongst all HCWs. The theme dealt with the benefit that the long-acting contraceptive implant afforded the user because it 1) it did not rely on frequent user adherence, 2) allowed for less pain due to reduced frequency of administration, and 3) improved user convenience |
| | Side Effects: A theme characterising the importance of side-effects amongst contraceptive implant users. Highlights side-effects as a key barrier to contraceptive implant acceptability. |
| | Migration of the Contraceptive Implant: the theme that deals with concerns over the migration of the implant away from the insertion site. Important theme as impacted the development of myths about the contraceptive implant. |
| | The Social Context: Important gatekeepers: Theme that highlights the importance of key stakeholders for contraceptive use acceptability amongst women. |
| Important Characteristics for an HIV Prevention Implant | Implant Size and Duration of drug efficacy: This theme deals with the preferred size and period of protection that HCWs feel would offer the highest acceptability amongst potential users. |
| | Palpability: this theme deals with the importance of palpability for improved ability to feel and remove the implant. Sub-themes of discretion concerns and issues of adherence and links to biodegradability are also included. |
| | Biodegradability of implant: this theme deals with the HCWs perception of biodegradability as a benefit to user acceptability. |
| Capacity Development of Service Providers to Provide Implants | This theme highlights the training and capacity development that HCWs received during the roll-out of the contraceptive implant and lessons we must learn for future training on new technologies such as the PrEP implant. |
| Engaging the Local Community on New Technologies | This theme highlights the importance of community engagement, the challenges and limitations of previous community engagement for the contraceptive implants. It highlights lessons learned for the future rollout of new technologies like the PrEP implant. |

Additionally, both CHWs and nurses alike emphasised that clinics were often understaffed and under-resourced. The long-acting nature of an implant was a benefit for reducing the burden on HCWs and on potential user's time:

"*Isn't some [users] are employed, so they will not have to frequently request day-offs causing employers to complain, and those that are studying don't have to miss any classes. So three years is very convenient, she would only return after three years to the clinic" (IDI13_ Nurse*)

Thus, the long-acting nature of an implant may have two-fold benefits; it reduces the work burden of the clinics and reduces the time burden on potential users. However, nurses highlighted that a PrEP implant would still impact clinic workloads as a user would still need to undergo regular HIV testing saying, "*A person [should] use the implant for at least a year and that person should get tested [for HIV] again and be found to have not been infected". (IDI13_ Nurse*)

**Side effects of the contraceptive implant.** Across all the interviews side effects were the primary critical barrier to the continued use of contraceptive implants. The side effects experienced appeared to have the greatest effect on long-term use or continued use of contraceptive implants. As one CHW reported:

*I know [two people] who stayed [adherent] even when they were having non-stop bleeding but they did stay for 3 years only [. . .], then said they [would] not re-insert after removing it (IDI19_CHW)*

One CHW who had used the contraceptive implant provided insight into the side effects she experienced. These motivated her early removal of the implant:

*I used the Implanon myself but it did not treat me well. I was dizzy, had headaches, pains and I had problems with heavy bleeding. I was bleeding and had excruciating pain- period pains that I had never had before. When I was about to menstruate it felt like a rock was going through me. (IDI15_CHW)*

Side effects may be enough to discourage use amongst women, and may become a reason why women choose to remove implants regardless of whether or not they experience the side effect themselves. As one nurse noted with regards to heavy-bleeding and the contraceptive implant:

*They complain of heavy bleeding but sometimes when we check some of them they use that as an excuse it is not that they are really having heavy bleeding, maybe they just don't want it. (IDI7_Nurse)*

The experience of side effects is critical to decision-making and acceptability. Side effects, experienced or a product of social learning (learning through observation), may serve to discourage use regardless of efficacy. In this context, the side effects were primarily the result of the active drug and not the device, and menstrual bleeding was a particular barrier to contraceptive implant uptake. The active drug in PrEP implants may not result in similar side effects, and side effects may not have the same salience in PrEP implants as they do with contraceptive implants. However, monitoring side-effect profiles throughout clinical PrEP implant development will be critical to future acceptability and should form a part of future community messaging development.

**Migration of the contraceptive implant.**   The movement of the contraceptive implant from its original position was a concern for users, as noted by a nurse:

*I have experienced it because so many ladies have come to remove the implants because they were not on the same side where they had been inserted initially. Normally there's a specific position in which you put them, but they would relocate, whilst others (implants) would relocate and they wouldn't feel it, then it would have to be removed in [surgical] theatre. (IDI4_Nurse)*

The potential for movement of the implant from the original insertion area becomes the basis for myths and concerns about the implant travelling into other organs [25,26] and highlights the importance of practical training on correct insertions procedures. The issue of migration of the implant was closely linked to the issue of palpability and biodegradability as well.

**The social context: Important gatekeepers affecting uptake of contraceptive implants.** The decision to use implants takes place within the broader social context in which a person lives. Both nurses and CHWs noted that acceptance by certain gatekeepers' (like partners and church membership) impacts whether a woman decides to use the contraceptive implant. Partners will affect implant uptake, a CHW noted that young women may "*end up being scared*

*that their partner will kill them because they have an implant and the partner wants a child."* *(IDI15_CHW)*. This highlights the potential for intimate partner violence (IPV), especially in the context where the partner may not be supportive. It shows that both social (i.e. relationships) and broader structural issues (i.e. female sexual and reproductive health and rights) may hinder uptake of sexual and reproductive health services. Other stakeholders such as churches may further influence product use, as noted by one nurse:

> *Some men were very strict and refused for their women to have an Implant inserted, particularly those that belong to [. . .] religion. They would say that it is not allowed in their religion and any form of contraception is not allowed. (IDI4_Nurse)*

Social influences, especially partners and important social institutions are powerful players in influencing the uptake and acceptability of sexual and reproductive health services and new HIV technologies amongst women. The effective rollout of new products requires the *continued* engagement of several different stakeholders as it will impact whether products are used.

## Important characteristics for an HIV prevention implant

HCWs had clear opinions about what they thought the implant for HIV prevention should look like to improve acceptability amongst potential users. Importantly, opinions were highly congruent across both nurses and CHWs. Broadly, HCWs spoke about three key areas: 1) Implant size and duration of drug efficacy, 2) Palpability and 3) Biodegradability of the Implant.

**Implant size and duration of drug efficacy.** Overall, most HCWs felt that the PrEP implant should be the same size as the current contraceptive implant or smaller. Some CHWs felt that a PrEP implant slightly larger than the contraceptive implant would be acceptable if this larger size coincided with a longer-acting product:

> *I think the big rod that lasts for [a] longer period will be much better, because as much as people come to clinics but they are sometimes reluctant because clinics are always full. (IDI17_CHW)*

All but one HCW felt that the implant should last for a minimum of one year, but two years or more was ideal to *"delay [users] coming to the clinic"*. One concern raised by a CHW highlighted that long-acting products could stop users from checking if the product was still active and working properly, and suggested that regular check-ups should be implemented saying *"It should not last for a long time, there should be a period where they can check if it is still working. . ." (IDI15_CHW)*.

The length of time the drug remained efficacious was not the only important factor. Perceived efficacy or lack thereof could be a barrier to uptake. Three HCWs relayed experiences where patients fell pregnant while using a contraceptive implant and this fuelled discouraging myths about the product and its efficacy:

> *I think the other thing that made others [potential users] not to be interested is that they have heard the others telling them that I [had the implant] inserted then after 3 months, I became pregnant. (IDI12_Nurse)*

This highlights important considerations for an HIV prevention implant. Careful messaging around how HIV prevention products reduce but do not eliminate the risk of HIV infection will be critical to ensure that users are not disappointed by their expectations of efficacy versus the actual efficacy of a product to prevent HIV infections.

**Palpability.**  Overall, most nurses and CHWs interviewed felt that the PrEP implant should be palpable so that it could be located and easily removed. Lessons from the contraceptive implant meant that HCWs felt that because palpability may be equated with safety (i.e. it had not migrated) and product efficacy (i.e. because if a user can feel it, they think it is working), young women would want to feel the implant. As one nurse said:

*I think [being] palpable will be good. [The] first reason why I'm saying that [is that] even the owner will be sure it is still there. Another reason [is that they] will feel that I'm still safe, because once it disappears even the patient will have that doubt, "Is it still working or is [it] going to damage one of my organs?" (IDI2_Nurse)*

However, one caution raised by CHWs was that palpability may negatively affect the discretion of the implant to safeguard young women against unwanted harassment and violence from partners. As learned from the use of contraceptive implants, a CHW noted:

*[. . .] I was in a festival and this boy said [that] they pretend to be playful with a girl by touching her arm, in order to confirm if she has the Implanon in her arm. My question is, is there a way for it to be placed in an area where a person will not be able to see it so that it can be hidden, for males not to know that there is an implant there. Because they will end up being scared that their partner will kill them because they have an implant and the partner wants a child. (IDI15_CHW)*

This highlights a potential disconnect between what is a benefit for an HCW versus a potential user. Being palpable may have benefits for HCWs as it eases the removal of an implant, but for discretion, palpability may serve as a challenge to uptake amongst potential users. It also highlights an area of potential user conflict where the need for discretion will be weighed against the belief that palpability equates with continued efficacy and as a means to confirm the implant has not migrated.

**Biodegradability of the implant.**  Perceptions of a biodegradable implant were largely positive from both nurses and CHWs and interlinked with the sub-theme of palpability. Most HCWs felt that a biodegradable implant would result in less pain as it would avoid the need for removal, and result in less scarring which has been a problem with the contraceptive implant. This would make the PrEP implant more acceptable:

*I: Which would be better for this HIV one–[. . .] would it better that they come remove it or that it is absorbable? Just like stitches are absorbable.*

*N: In my opinion, I think it should be absorbable because it is easy to do the insertion, but when removing it, [that is] when it results in scarring which no one likes. (IDI13_ Nurse)*

An area of ambivalence was the impact that a biodegradable PrEP implant could have on adherence. Supporting adherence, a CHW noted that the disappearance of the PrEP implant may serve as a reminder to have it replaced, as the user would equate the disappearance with reduced efficacy and hence the need to have it replaced: "*I think it's better to be biodegradable because if I have [to] remove it at the clinic, I will have this [implant efficacy] in my heart that it is still effective because I can feel it's there*"(IDI14_CHW). However, this same benefit could serve to negatively affect adherence and ultimately its prevention effectiveness because "*[removing an implant at a] clinic [may remind users] of the duration of the implant because sometimes people forget their regular visits, whereas when it is biodegradable, they would not bother going to the clinic to get another one*" (IDI9_CHW).

HCWs raised that biodegradability may raise concerns about where the implant goes when it biodegrades. One nurse switches her answer when contextualising the implementation of a biodegradable implant in her community, based on her experiences providing contraceptive implants:

*I think absorbable is ideal. [. . .]But maybe they will ask—the client will ask about where it is going when it is absorbed because, they are reluctant to most of the things [. . .] even with the no period thing of Implanon, they are curious about where the blood is going if you do not see your periods. So [being] absorbable will raise questions with the community. It is better to remove it so we can show it to them (IDI16_Nurse)*

These concerns highlight that while biodegradability may have benefits for HCWs who have high workloads; this option may have a complex impact on community acceptability.

## Capacity development of service providers to provide implants

The continued capacity development of HCWs to provide information about and then implement new technologies is critical. Many nurses and CHWs reported that they had received at least once-off training, however, only nurses reported receiving formal and limited mentoring. As one nurse stated:

*I: And eh, have you received any mentorship?*

*N: Yes.*

*I: Yes. From who?*

*N: From (Clinician A), because when I came back from the training I started having problems because I was not used [to it], I was not skilled enough to do it.*

*I: Okay. And how many days or how many hours?*

*N: It was, no, it was. . . maybe three hours. (IDI16_Nurse)*

In most instances, training did not happen for all nurses. Rather a train-the-trainer system whereby training was done for a single nurse or a couple of nurses, who then trained the remainder of the clinic staff was implemented. This causes problems for long-term service provision when staff rotation or attrition occurs, and new staff need to be trained:

*I: Did all the nurses do the Implanon training?*

*N: No.*

*I: How were you chosen to be part of the implant insertion training?*

*N: The ones that were trained previously by another facility [training session] left the clinic. The other one is new and I think it depended on one's interest because some did not even attend the briefing that was done by the nurse who attended the training. (IDI7_Nurse)*

Training appeared to be inconsistent for CHWs. The quality of the CHW training appeared dependent on the supervisor who was responsible for the CHWs and could vary from a five-day-long training to no training at all. As noted by a CHW:

*C: No I haven't attended any training. I am using the information that I have overheard in the clinics.*

*Interviewer*: *Okay. So the nurses didn't give you any training so that you will be able to inform the people well about the implant*?

*C*: *No. We depend on the information that we have overheard about the implant.* *(IDI11_CHW)*

Training and plans for continued training of staff, especially those that interface directly with potential users, will be critical to the rollout of future PrEP implants.

## Engaging the local community on new technologies

Similar to the inconsistencies in the level and quality of training, community engagement activities at the start of new campaigns seemed to depend on the clinic and the resources available to them. In the first extract the nurse highlights that the clinic engagement occurred through multiple teams:

*I*: *The community where you were working at [community name], was the community ever engaged before this method [Contraceptive Implant] was introduced to say this what we'll be introducing just to engage the community*?

*N*: *Yes, because we normally do health education in the mornings, I would say yes. Because there are also school nurses that do that and the school health team.*

*I*: *They go out to the community*?

*N*: *Yes, yes.*

*I*: *How do they do it, do they go house to house?*

*N*: *Family health go[es] house to house, school nurses' school to school, mobile [clinics] go to certain points and us at the clinic. (IDI4_Nurse)*

Then, a CHW at another clinic highlighted that outreach was limited to talks in the clinic only, with no outreach in the community:

*C*: *The community was engaged because it was also announced on radio and on television during the NEWS time. The counsellors did educate the people who came to the clinic in the morning about this contraceptive implant.*

*I*: *So there was no outreach campaign on implants*?

*C*: *Not that I know of. (IDI9_CHW)*

Some HCWs reported that with the rollout of PrEP implants, there should be improved coordination amongst different stakeholders to ensure that engagement plans are coordinated and that information is consistent. A CHW highlighted *"I would like the Department of Health to involve all health employees and NGOs in order to discuss a way forward. After that, when they have decided that it will be introduced [we can move forward]"(IDI15_CHW)*. In addition, a nurse interviewed highlighted the importance of integrating services into schools to target young people, highlighting that:

*I think it would be good if we were to go to schools for the [contraceptive] insertions because the younger ones do not want to come to the clinic. They are the ones who are the target. It is difficult for them to come to the clinic because they come out of school late. They come to the*

*clinic late and then they have to leave late or else they have to start here and delay going to classes. (IDI16_Nurse)*

The findings highlight the inconsistency in engagement plans with potential users and the communities in which the user will use them. Systematic, sustainable programmes are needed to ensure long-term community engagement with new HIV prevention technologies.

## Discussion

Young women in sub-Saharan Africa remain the most vulnerable to HIV infection and expanding our toolbox of HIV prevention options is critical to achieving epidemic control. Health care workers play an important role in ensuring that efficacious health products are accessed and utilised by potential beneficiaries and end-users [17]. They offer an invaluable perspective on how communities will respond to the introduction of new technologies and should play an important role in early clinical product development, and driving the acceptability and uptake of these products as they become available. Lessons from the rollout of contraceptive implants provide important insights into the factors that may impact the acceptability of PrEP implants.

Likely, many of the benefits and challenges that characterise the contraceptive implant will similarly affect PrEP implants [7,17,27,28]. For HCWs, especially the nurses, the benefits of a long-acting PrEP implant were explicitly linked to its impact on the running of the clinic and managing workloads: products that reduce visit frequency and are palpable and flexible to make removal easier. However, these preferences highlight complex issues that go beyond a tick-box of key characteristics that a PrEP implant should have, and highlight that tensions may exist between the preferences of HCWs and how this translates to impact potential users.

Similar to research in other parts of South Africa [17,29], and consistent with the benefits of contraceptive implants, both nurses and CHWs had a preference for long-acting products. However, while having a long-acting product (> 1 year) was preferred and considered a major benefit of the current contraceptive implant, for HIV prevention, the implant may not reduce workloads as significantly as hoped. For HIV prevention, long-acting products will not negate the need for regular HIV testing to ensure that a client has not seroconverted, and is timeously referred to HIV treatment services. Thus, the introduction of these technologies is likely to require clients to still come to clinics regularly and as the implant will likely be available to both men and women it may increase the numbers seeking these services. Further, while long-acting products offer long-term protection, user expectations of protection against HIV infection will need to be managed. Limits to how efficacious a product needs to be easily and carefully explained to ensure that users are not disappointed by their expectations of efficacy versus the actual efficacy of a product to prevent HIV infections.

Palpability was considered important to both nurses and CHWs for different reasons. Nurses felt that palpable PrEP implants made it easier to locate and remove, while CHWs felt it would help users be certain that the implant had not moved, and that users would associate its presence with the fact that it was still working. On the negative side, palpability may have the unintended consequence of reducing discretion and invisibility with partners or family members, a major benefit of injectable long-acting HIV prevention options [17,30]. As partners have a significant influence on the use of current HIV prevention technologies [31,32], users of PrEP implants will weigh concerns about discretion against the perceived benefits of the implant to reduce HIV infection, mediated by their risk perception. So, while palpability may serve to make insertion and removal easier for nurses, it may negatively influence the acceptability amongst users, especially female users. Consideration of where an implant is

placed on the body and how social relationships will affect sexual health decisions will be critical. An issue that requires exploration with potential users is whether using the PrEP implant could become a "badge of honour", being equated with an HIV negative status. However, there is the risk that partners may still assume the PrEP implant is a contraceptive implant which could cause problems if the partner wants to have children. Careful messaging designed to explain the differences between the contraceptive and PrEP implant will need to be developed to ensure that the challenges associated with one product are not conflated with the other. Partners may play an important role in influencing the uptake of new technologies as seen with contraceptive implants. Use of a PrEP implant without the knowledge of a partner could be interpreted as an indicator of a lack of trust, or as "proof" of infidelity by the female partner and may increase her risk of intimate partner violence. More research about how partners and family members will influence the uptake of PrEP implants is needed to understand the impact these important social connections will have on uptake, especially if PrEP implant discretion is limited.

Biodegradability had similar important implications for acceptability. Our data confirm previous research, which found that most HCWs (both nurses and CHWs) are in favour of an implant that would biodegrade [17]. The motivation for biodegradability was linked to reducing workloads, a lack of capacity to remove implants and minimising the pain associated with removal for users [17,33]. However, biodegradability may give rise to community myths similar to those found in the contraceptive literature [25,34]. A lack of understanding about where the product disappears to, concerns about how it may affect the body and whether other potential unrelated health issues will be attributed to the disappearance of the implant will all serve to undermine potential acceptability regardless of its benefits for the health care system. HCWs also raised concerns that the "disappearance" of the product may be associated with reduced palpability and hence perceived efficacy which may lead to concerns amongst potential users that they are no longer protected. While some felt that this may positively impact adherence (reminding users to get a new one) others felt that the slow biodegrading implant could negatively impact adherence because users would not have the reminder of a palpable implant in their arm. This raises an opportunity for modifiable characteristics of implant technologies within the clinical development phase. It highlights that within the development of the same products, broadening the range of the product characteristics available may serve to help fit the product to the needs and preferences of the user more specifically.

An important lesson from the roll-out of the contraceptive implant was that side effects (especially heavy bleeding) were a key reason for poor user uptake and early removal which may in turn fuel myths and social learning which deter other potential users [17,28,31]. While side effects may be attributed to the active drug used in the implant rather than the delivery method, careful consideration of the side-effect profile on PrEP implants will be critical to real-world usability. Negative side-effects have been a key factor impacting adherence to ARV medication [35] as well as other available HIV prevention technologies, like oral PrEP [29]. Therefore community education about the differences between the contraceptive implant and PrEP implant will be critical to avoid negative associations with the contraceptive implant being conflated with those of the PrEP implant. Careful, and transparent disclosure about possible side effects and how these differ from other available HIV prevention technologies will be an important part of the messaging and community engagement processes that need to occur. The experience of side effects will likely be critical to the decision-making of users regardless of product efficacy.

Training and community engagement form the pillars of developing sustainable healthcare systems that can respond to the needs of communities. Similar to contraceptive implants, the capacity of HCWs and lack of long-term sustainable plans for practical training and retraining

staff (especially on insertions and removals) may negatively impact PrEP implant provision [7,17,36]. Sustainable models for mentorship and practical training of HCWs will be needed to ensure the successful rollout of new technologies. In addition, multi-pronged, high quality, and sustained community engagement will be critical to the rollout of PrEP implants. Without empowered and knowledgeable providers and communities, or sustained programmes for engagement, it will be easy for misinformation to spread, and for the initial excitement for new products to fade. HIV prevention technologies, like the PrEP implant, may benefit from the contraceptive implant in terms of acceptability because it will be available to both men and women. The engagement should be broad-based, targeting multiple stakeholders including male and female partners, parents, and social stakeholders such as schools and religious institutions. While using a PrEP implant may be inherently an individual choice, the decision to use it will be influenced by a complex interplay of other factors such as community acceptance, religious affiliation, and partner approval; and broad-based community engagement and promotion of use amongst both men and women should be done from the start. While a PrEP implant may not be acceptable for all users, any long-term acceptability will need to target multiple stakeholders to ensure that the product does not become irrelevant because a user is unable to overcome the various challenges to safe and discrete use in the real world.

Our study has some limitations. The sample size is small and situated in a specific rural geographic location, limiting the representativeness and generalisability of our findings. However, given the diversity of epidemics within and between countries a localised understanding is critical to know your epidemic and breaking chains of transmission. The sample also consisted of all cis-gender women (at or just past reproductive age) and so some of their responses may be biased by their personal experiences of having used contraceptive implants. The findings are consistent with other studies conducted in different parts of South Africa [7,17,37]. The findings are formative and need to look at how HCW opinions will differ from that of potential users. Research investigating how these findings may be different for potential users is underway. Additionally, research on community engagement strategies and the development of lexicons for promoting the use of new technologies amongst different age groups and user groups will be needed.

When we consider what previous research has suggested are the preferred modifiable characteristics of PrEP implants by potential users, our findings highlight that there may be tension between the characteristics of PrEP implants that are perceived to be beneficial by HCWs and by potential users. These provider identified characteristics have to be carefully considered and weighed up against potential user preferences and may need to be addressed in provider training to introduce new technologies. Finding a balance between the needs of HCWs that accommodate their heavy workloads, limited resources at points of delivery of care and the needs and preferences of potential users need to be carefully considered. These data highlight the need for sustained and multi-pronged approaches to introducing new health technologies into communities. Complementary social science research, occurring alongside the development of new products may facilitate effective products to reach their full potential for HIV prevention.

## Supporting information

**S1 Appendix. The in-depth Interview Guides for the Nurses and Community Health Care workers.**
(PDF)

## Acknowledgments

The authorship team would like to thank all the study staff, the CAPRISA Vulindlela Community Research Support Group, the Vulindlela community, Provincial and District Departments of Health, and the healthcare workers willing to contribute to and participate in the study. We would also like to acknowledge Brett Marshall and Revina Munsamy, two Research Psychology masters fellows who assisted the team with coding the data.

## Author Contributions

**Conceptualization:** Hilton Humphries, Michele Upfold, Gethwana Mahlase, Makhosazana Mdladla, Tanuja N. Gengiah, Quarraisha Abdool Karim.

**Data curation:** Hilton Humphries, Michele Upfold, Gethwana Mahlase, Makhosazana Mdladla.

**Formal analysis:** Hilton Humphries, Michele Upfold.

**Funding acquisition:** Hilton Humphries, Tanuja N. Gengiah, Quarraisha Abdool Karim.

**Investigation:** Hilton Humphries, Michele Upfold, Quarraisha Abdool Karim.

**Methodology:** Hilton Humphries, Michele Upfold, Gethwana Mahlase, Tanuja N. Gengiah, Quarraisha Abdool Karim.

**Project administration:** Hilton Humphries, Michele Upfold, Gethwana Mahlase, Makhosazana Mdladla.

**Resources:** Hilton Humphries, Michele Upfold.

**Software:** Hilton Humphries, Michele Upfold.

**Supervision:** Hilton Humphries, Michele Upfold, Gethwana Mahlase, Makhosazana Mdladla.

**Validation:** Hilton Humphries, Michele Upfold.

**Visualization:** Hilton Humphries, Michele Upfold.

**Writing – original draft:** Hilton Humphries.

**Writing – review & editing:** Hilton Humphries, Michele Upfold, Gethwana Mahlase, Makhosazana Mdladla, Tanuja N. Gengiah, Quarraisha Abdool Karim.

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
