## [Decision Letter · Decision Letter 0]

3 Nov 2021

PONE-D-21-27516Implants for HIV prevention in young women: Provider perceptions and lessons learnt from contraceptive implant provisionPLOS ONE

Dear Dr. Humphries,

Thank you for submitting your manuscript to PLOS ONE. After careful consideration, we feel that it has merit but does not fully meet PLOS ONE’s publication criteria as it currently stands. Therefore, we invite you to submit a revised version of the manuscript that addresses the points raised during the review process.

We look forward to receiving your revised manuscript.

Kind regards,

Rupa R. Patel, MD

Academic Editor

PLOS ONE

Journal Requirements:

2. When reporting the results of qualitative research, we suggest consulting the COREQ guidelines  or other relevant checklists listed by the Equator Network, such as the SRQR, to ensure complete reporting (http://journals.plos.org/plosone/s/submission-guidelines#loc-qualitative-research). Moreover, please provide the interview guide used as a Supplementary File.

"The authorship team would like to thank all the study staff, the CAPRISA Vulindlela Community Research Support Group, the Vulindlela community, Provincial and District Departments of Health, and the healthcare workers willingness to contribute to and participate in the study. We thank the South African Medical Research Council Special initiative grant (00251), South African Department of Science and Innovation-National Research Foundation Centre of Excellence in HIV Prevention for their funding support. We would also like to acknowledge Brett Marshall and Revina Munsamy, two masters fellows who assisted the team with coding the data"

"Funding: Funding for the study was provided by South African Medical Research Council Special initiative grant (00251), and partial support from the Department of Science and Innovation-National Research Foundation Centre of Excellence in HIV Prevention"

Editor Comments:

This you for this manuscript on a much needed topic to enhance global PrEP rollout.

Please find my comments below:

Overall formatting

-I agree with the comments provided by the reviewers on grammar, capitalization of headings, and consistency regarding the verb tense throughout the manuscript.

Introduction

Methods

-Under procedure: Please clearly state the study inclusion criteria for example, age, place of work, language, able to consent, gender, education level, experience with contraception discussions, experience # of years of work, type of professional, experience with implanting devices, experience with a certain # of contraceptives, etc.

-Under procedure: Please state if participants offered compensation for transportation if they were.

-Under procedure: Please provide more information on translation and back translation; was the guide created in English, translated into isiZulu and then back translated for accuracy?

-Under procedure: Please better describe the recruitment process and if there were study marketing materials used to recruit potential participants.

-Under procedure: Was there purposeful sampling to recruit those with experience with placing contraceptive devices and those who had no experience?

-Was there a survey that was administered, in addition to the qualitative guide, to capture demographics, years of experience, and other information presented in Table 1 and that you gathered but did not present in this paper? If so, please include it in the appendix as well as describe the survey’s domains or info in the text in the methods section.

Results

Table

-Please create a more tradition Table 1 that includes N=18. Please use the headings in a column to the left and write them as rows. Then supply columns for CHW and PHCs and Both CHW/PHCs. You can place the Ns in the heading rows.

-In the table, please include a row for age (range and median or mean) and a row for years of experience in the public health settings (range and median or mean).

Quotes

-For the labels for the quotes, please stay consistent with PHC Nurse or CHW. Nurse is labeled as Nurse.

-Please label all the quotes. The last quote in the long-acting nature section has no participant label.

-If possible, please add more CHW quotes in the long acting section, if applicable.

Supplemental materials

-Please provide the interview guide in English and in isiZulu.

-Please provide the survey administered to capture demographics etc both in English and in isiZulu. This would be different than the qualitative questions.

-Please provide a visual of any frameworks/theories that the qualitative guide and this study was based on, if any.

Reviewers' comments:

Reviewer's Responses to Questions

**Comments to the Author**

1. Is the manuscript technically sound, and do the data support the conclusions?

Reviewer #1: Yes

Reviewer #2: Yes

2. Has the statistical analysis been performed appropriately and rigorously? 

Reviewer #1: N/A

Reviewer #2: N/A

3. Have the authors made all data underlying the findings in their manuscript fully available?

Reviewer #1: Yes

Reviewer #2: No

4. Is the manuscript presented in an intelligible fashion and written in standard English?

Reviewer #1: Yes

Reviewer #2: No

5. Review Comments to the Author

Reviewer #1: I made minimal revisions related to grammar and format. I recommend special attention to capitalization of the headings and subheadings, use of full name before use of acronyms, and tense (learnt vs. learned, for example). I highlighted in blue an area that I suggest could be more fully examined and discussed. For example the theme termed as social context also points to intimate partner violence (IPV). Beyond social determinant, IPV is related to structural level determinants of health (i.e. policy and laws) that ultimately impact the individual in their utilization of a desired implant method. My other major recommendation is related to the possible limitations of the findings = as the sample may be biased due to their *own* (personal) experiences of implants as a contraceptive method for pregnancy prevention, given that the respondents are all women and may most likely be within their reproductive years or recently past this period in their own lives. Finally, I recommend a statement early on in the manuscript that within the context of this study, the authors/researchers are identifying gender - "women" - defined as cisgender women (i.e. individual assigned female sex at birth that identifies their gender as woman).

Reviewer #2: Thank you for this important study. Your manuscript highlights the importance of including stakeholders and end-user service providers in the design and delivery stages of HIV Prevention technologies, particularly areas with a high HIV burden.

MINOR Recommendations: Please copyedit the manuscript. Particularly cross-check sentence cases, headers, and tables to ensure they are consistent with the required formatting guidelines.

MAJOR Recommendations: More clarity regarding the primary perspective/experience being explored.

The introduction states the importance of exploring both provider and end-user engagement however the study sample does not include end-users.

The manuscript aims to explore the service provider perspectives of delivery of PrEP implants; however, there are findings and exemplar quotations that support the perspectives of health care workers as potential users of PrEP implants for themselves as women.

For example the quotation: "I see a lot of benefits, mainly time-saving, as I don’t need to [come] to the clinic all the

time. It saves both time and money. Also, if I feel pain I would feel it once and the next

time I feel pain will be after three years when the implant is removed unlike feeling pain

every 3 months. (IDI3 – Nurse)"

It seems to reflect the Nurse's perspective of using the PrEP implant as opposed to the Nurse's perspective of providing the care, service, or anticipated attitudes from potential users.

In the conclusion you state, "Our findings highlight that the characteristics of PrEP implants that are perceived to be

beneficial by HCWs may not align with that of potential users."

I am unable to find within the manuscript any interview data/findings from potential users as a comparison to firmly support this statement.

I hope you all find these recommendations helpful. Thank you for your excellent contributions to the field.

6. PLOS authors have the option to publish the peer review history of their article (what does this mean?). If published, this will include your full peer review and any attached files.

Reviewer #1: **Yes: **Gabriela Santana Betancourt

Reviewer #2: No

---

## [Author Response · Author response to Decision Letter 0]

15 Dec 2021

Comment: Ensure that the manuscript meets the PLOS One style requirements including those for file naming. Response: Thank you, we have reviewed and confirmed that the manuscript meets the style guidelines of the journal.

Comment: Consult the COREQ guidelines for qualitative research. Response: The COREQ were consulted and the reporting meets the requirements of the guidelines. The interview guide has been provided as part of the supplementary data for the manuscript.

Comment: Funding Statement. Response: Funding statement should read: Funding: Funding for the study was provided by South African Medical Research Council Special initiative grant (00251), and partial support from the Department of Science and Innovation-National Research Foundation Centre of Excellence in HIV Prevention. Acknowledgements now reads: The authorship team would like to thank all the study staff, the CAPRISA Vulindlela Community Research Support Group, the Vulindlela community, Provincial and District Departments of Health, and the healthcare workers willing to contribute to and participate in the study. We would also like to acknowledge Brett Marshall and Revina Munsamy, two master’s fellows who assisted the team with coding the data. 

Comment: Please clearly state the study inclusion criteria for example, age, place of work, language, able to consent, gender, education level, experience with contraception discussions, experience # of years of work, type of professional, experience with implanting devices, experience with a certain # of contraceptives, etc. Response: Thank you for the comment. This has been updated in the procedures section. The following has been added: To be eligible for the study, the participant had to be a professional nurse or CHW working at a primary health care clinic, and have self-reported experience with providing information about Implanon (CHWs and nurses) or contraceptive services (nurses) which included Implanon. There was no upper age limit or gender criteria for participating in the study.

Comment: Please state if participants offered compensation for transportation if they were. Response: Thank you for the comment, this was provided in the procedures. We have added additional text to ensure it is clear. To minimise disruption, interviews were conducted in a private space at the PHC clinic, and each interview lasted approximately one hour. All IDIs were conducted in isiZulu and were audio-recorded, transcribed and then translated into English. English transcripts were quality checked by a research assistant fluent in isiZulu and English against the isiZulu interview recordings for accuracy. Participants were offered refreshments but no reimbursement for their time or travel.

Comment: Please provide more information on translation and back translation; was the guide created in English, translated into isiZulu and then back translated for accuracy? Response: Thank you for the comment. We have added additional clarity. The semi-structured interview guide was created in English. The experienced interviewer, who is also an investigator on the study, a trained nurse, and fluent in English and isiZulu, was involved in developing the semi-structured interview guide, and translated the questions into isiZulu during the interview process. All transcripts underwent quality assurance processes to check the transcription against the audio, and the guide and to ensure the translation was correct. Corrections were made and translations finalised 

Comment: Please better describe the recruitment process and if there were study marketing materials used to recruit potential participants. Response: Thank you for this comment. We have added the following text: To recruit participants, our CAPRISA community team approached each of the clinics and asked the clinic to identify the nurses and CHWs responsible for providing or doing community outreach with regards to family planning services at the clinic. Those who met the eligibility requirements were asked to participate in the study.

Comment: Was there purposeful sampling to recruit those with experience with placing contraceptive devices and those who had no experience? Response: This has been clarified with the inclusion criteria. The nurses had to have experience providing information on family planning, as well as providing family planning services.

Comment: Was there a survey that was administered, in addition to the qualitative guide, to capture demographics, years of experience, and other information presented in Table 1 and that you gathered but did not present in this paper? If so, please include it in the appendix as well as describe the survey’s domains or info in the text in the methods section. Response: Thank you for highlighting this omission. We have added the following text: Following written consent, a short interviewer-administered survey was completed to capture basic demographics, experience with Implanon, and qualification, after which an individual in-depth interviews (IDI) using a semi-structured interview guide were completed at a time convenient to the participant. The survey is included as part of the interview guide.

Comment: Please create a more tradition Table 1 that includes N=18. Please use the headings in a column to the left and write them as rows. Then supply columns for CHW and PHCs and Both CHW/PHCs. You can place the Ns in the heading rows. In the table, please include a row for age (range and median or mean) and a row for years of experience in the public health settings (range and median or mean). Response: Thank you for the comment, the table has been updated as requested. We only asked nurses about experience with Implanon, as this was most relevant to the study purpose. All relevant information is reflected in the table.

Comment: -For the labels for the quotes, please stay consistent with PHC Nurse or CHW. Nurse is labelled as Nurse.-Please label all the quotes. The last quote in the long-acting nature section has no participant label.-If possible, please add more CHW quotes in the long acting section, if applicable. Response: They have been updated to read Nurse, and made sure all quotes have a label. An additional two quotes have been added to the long acting section, one from a nurse, and one from a CHW. These read: I: Okay. Okay. In your contraceptive counselling, eh…what do you cover?N: Okay, firstly, I cover the advantages because one of the advantages of the Implanon, is that it stays long. We are using that three-year one, so that is the advantage is that they do not have to come to the clinic every three months to access family planning. Another advantage is that if you feel like falling pregnant you can come to the clinic and remove it and go back to your fertility. The other advantage is that eh…there is no need to tell your partner about it (IDI16_Nurse)I: Okay. What do you like about the implant? C: It stays for a long time. It’s unlike the injection that you have to inject every 2 months. It stays for up to 3 years. And you can replace it when it time has lapsed. (IDI11_CHW)

Comment: -Please provide the interview guide in English and in isiZulu.-Please provide the survey administered to capture demographics etc both in English and in isiZulu. This would be different than the qualitative questions.-Please provide a visual of any frameworks/theories that the qualitative guide and this study was based on, if any. Response: Thank you for the feedback. The guides are included, the survey administered is included as part of these guides. There was no framework, as the analysis looked at previous experiences to develop preliminary codes, looking for feedback on user preferences, challenges and barriers as highlighted by the HCWs. This is explained in the analysis section.

Comment: I made minimal revisions related to grammar and format. I recommend special attention to capitalization of the headings and subheadings, use of full name before use of acronyms, and tense (learnt vs. learned, for example). I highlighted in blue an area that I suggest could be more fully examined and discussed. For example the theme termed as social context also points to intimate partner violence (IPV). Beyond social determinant, IPV is related to structural level determinants of health (i.e. policy and laws) that ultimately impact the individual in their utilization of a desired implant method. Response: Thank you for the very helpful suggestions and comments. We have made the necessary changes. The issue of IPV is absolutely correct, and agreed with. We have highlighted this point in the results as: This highlights the potential for intimate partner violence (IPV), especially in context where the partner may not be supportive. This highlights both social (i.e. relationships) and broader structural issues (i.e. female sexual and reproductive health and rights) that may hinder uptake of sexual and reproductive health services. In the discussion we have added:Partners may play an important role in influencing the uptake of new technologies as seen with contraceptive implants. Use of a PrEP implant without the knowledge a partner could be interpreted as an indicator of a lack of trust, or as “proof” of infidelity by the female partner and may increase her risk of intimate partner violence.

Comment: My other major recommendation is related to the possible limitations of the findings = as the sample may be biased due to their *own* (personal) experiences of implants as a contraceptive method for pregnancy prevention, given that the respondents are all women and may most likely be within their reproductive years or recently past this period in their own lives. Finally, I recommend a statement early on in the manuscript that within the context of this study, the authors/researchers are identifying gender - "women" - defined as cisgender women (i.e. individual assigned female sex at birth that identifies their gender as woman). Response: Thank you for the helpful comments. We have added the following to the limitations section: The sample also consisted of all cis-gender women (at or just past reproductive age) and so some of their responses may be biased by their own personal experiences of having used contraceptive implants. In addition, we have added a cis-gender women definition in the demographics section.

Comment: Please copyedit the manuscript. Particularly cross-check sentence cases, headers, and tables to ensure they are consistent with the required formatting guidelines. Response: Thank you. We have checked the manuscript.

Comment: The introduction states the importance of exploring both provider and end-user engagement however the study sample does not include end-users. Response: Thank you for the comment. The paragraph starts by saying engaging potential users and service providers is important but then does state that this manuscript describes perspectives of HCWs specifically. 

Comment: The manuscript aims to explore the service provider perspectives of delivery of PrEP implants; however, there are findings and exemplar quotations that support the perspectives of health care workers as potential users of PrEP implants for themselves as women.For example the quotation: "I see a lot of benefits, mainly time-saving, as I don’t need to [come] to the clinic all the time. It saves both time and money. Also, if I feel pain I would feel it once and the next time I feel pain will be after three years when the implant is removed unlike feeling pain every 3 months. (IDI3 – Nurse)" It seems to reflect the Nurse's perspective of using the PrEP implant as opposed to the Nurse's perspective of providing the care, service, or anticipated attitudes from potential users. Response: Thank you for the comment. The question that was asked by the interviewer was “According to you as a health care professional, what are the benefits of an implant?” Therefore, the nurse uses “I” in her response, but she is using this vernacular to discuss her feelings about what she feels the benefits would be in general, and that these would therefore be benefits for others. This still conveys her beliefs and perspectives about how it would be a benefit for a potential user and is congruent with the aims of the paper. We have however added the following to the limitation section: The sample also consisted of all cis-gender women (at or just past reproductive age) and so some of their responses may be biased by their own personal experiences of having used contraceptive implants.

Comment: In the conclusion you state, "Our findings highlight that the characteristics of PrEP implants that are perceived to be beneficial by HCWs may not align with that of potential users." I am unable to find within the manuscript any interview data/findings from potential users as a comparison to firmly support this statement. Response: Thank you for highlighting the ambiguity in this sentence. We have updated to better reflect what we are saying i.e. that what previous research suggested are the user preferred characteristics of a PrEP implant may not align with that of providers. We have reworded to say: When we consider what previous research has suggested are the preferred modifiable characteristics of PrEP implants by potential users, our findings highlight that there may be tension between the characteristics of PrEP implants that are perceived to be beneficial by HCWs and by potential users.

---

## [Editor Report · Decision Letter 1]

16 Dec 2021

Implants for HIV prevention in young women: Provider perceptions and lessons learned from contraceptive implant provision

PONE-D-21-27516R1

Dear Dr. Humphries,

We’re pleased to inform you that your manuscript has been judged scientifically suitable for publication and will be formally accepted for publication once it meets all outstanding technical requirements.

Kind regards,

Rupa R. Patel, MD

Academic Editor

PLOS ONE

---

## [Editor Report · Acceptance letter]

5 Jan 2022

PONE-D-21-27516R1 

Implants for HIV prevention in young women: Provider perceptions and lessons learned from contraceptive implant provision 

Dear Dr. Humphries:

I'm pleased to inform you that your manuscript has been deemed suitable for publication in PLOS ONE. Congratulations! Your manuscript is now with our production department. 

Kind regards, 

on behalf of

Dr. Rupa R. Patel 

Academic Editor

PLOS ONE